# Manifold-aware Training: Increase Adversarial Robustness with Feature Clustering

## Abstract

The problem of defending against adversarial attacks has attracted increasing attention in recent years. While various types of defense methods (*e.g.*, adversarial training, detection and rejection, and recovery) were proven empirically to bring robustness to the network, their weakness was shown by later works. Inspired by the observation from the distribution properties of the features extracted by the CNNs in the feature space and their link to robustness, this work designs a novel training process called Manifold-Aware Training (MAT), which forces CNNs to learn compact features to increase robustness. The effectiveness of the proposed method is evaluated via comparisons with existing defense mechanisms, *i.e.*, the TRADES algorithm, which has been recognized as a representative state-of-the-art technology, and the MMC method, which also aims to learn compact features. Further verification is also conducted using the attack adaptive to our method. Experimental results show that MAT-trained CNNs exhibit significantly higher performance than state-of-the-art robustness.

## 1 Introduction

### 1.1 Background

Convolutional neural networks (CNNs) are increasingly used in recent years due to their high adaptivity and flexibility. However, Szegedy et al. (2014) discovered that by maximizing the loss of a CNN model w.r.t the input data, one can find a small and imperceptible perturbation which causes misclassification errors of the CNN. The proposed method for constructing such perturbations was designated as the Fast Gradient Sign Method (FGSM), while the corrupted data (with perturbation) were referred to as adversarial examples. Since that time, many algorithms for constructing such perturbations have been proposed, where these algorithms are referred to generally as adversarial attack methods (*e.g.*, Madry et al. (2018); Carlini & Wagner (2017); Rony et al. (2019); Brendel et al. (2018); Chen et al. (2017); Alzantot et al. (2019); Ru et al. (2020); Al-Dujaili & O'Reilly (2020)). Among them, Projected Gradient Descent (PGD) (Kurakin et al., 2017) and Carlini & Wagner (C&W) (Carlini & Wagner, 2017) attacks are the most widely-used methods. Specifically, PGD is a multi-step variant of FGSM, which exhibits higher attack success rate, and C&W attack leverages an objective function designed to jointly minimize the perturbation norm and likelihood of the input being correctly classified. The existence of adversarial examples implies an underlying inconsistency between the decision-making processes of CNN models and humans, and can be catastrophic in life-and-death applications, such as automated vehicles or medical diagnosis systems, in which unpredictable noise may cause the CNN to misclassify the inputs.

Various countermeasures for thwarting adversarial attacks, known as adversarial defenses, have been proposed. One of the most common forms in adversarial defense is to augment the training dataset with adversarial examples so as to increase the generalization ability of the CNN toward these patterns. Such a technique is known as adversarial training (Shaham et al., 2015; Zhang et al., 2019; Wang et al., 2020). However, while such methods can achieve state-of-the-art robustness in terms of robust accuracy (*i.e.*, the accuracy of adversarial examples), training robust classifiers is a non-trivial task. For example, Nakkiran (2019) showed that significantly higher model capacity is required for robust training, while Schmidt et al. (2018) proved that robust training requires a significantly larger number of data instances than natural training.

## 1.2 Motivation and Contributions

The study commences by observing the distribution properties of the features learned by existing training methods in order to obtain a better understanding of the adversarial example problem. In particular, the t-SNE dimension reduction method (van der Maaten & Hinton, 2008) is used to visualize the extracted features, as illustrated in Figures 1. The observation results reveal the following feature distribution properties:

- Non-clustering: same-class features are not always clustered together (*i.e.*, some points leave the clusters of their respective colors in Figure 1), which is at odds with intuition, which expects that the representative (for classification) features of same-class samples should be similar to one another.

- Confusing-distance: closeness between samples in the feature space does not imply resemblance of their prediction (especially for adversarial examples, as there are many triangles colored differently than surrounding points in Figure 1 (b)).

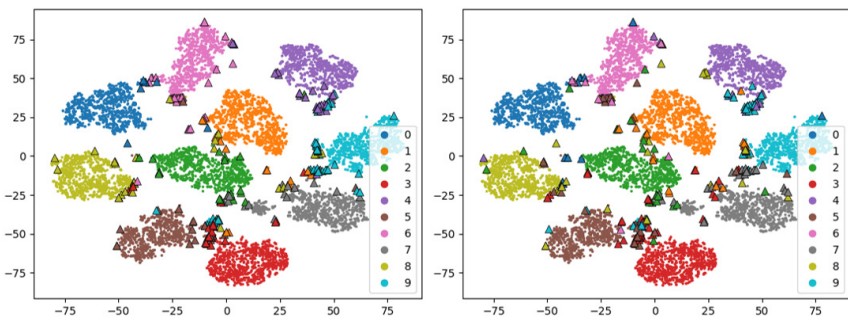

Figure 1: Features learned by models trained for MNIST using TRADES (Zhang et al., 2019) training, where the clean samples (points) are colored by predicted classes and the adversarial examples (triangles) are colored by (a) true labels or (b) predicted classes.

We confirm the validity of these observations through a numerical analysis of the matching rate between the closest cluster dominant class and the predicted class. Clustering analysis algorithms (*e.g.*, Ward's Hierarchical Clustering algorithm (Ward, 1963)) are leveraged to find the clusters formed by the CNN-learned features. Additionally, the match rate is defined as $\mathbb{E}_{\boldsymbol{x}}[\mathbf{1}_{\mathrm{dom}(\mathbb{C}^{(\boldsymbol{x})})=f(\boldsymbol{x})}]$, where $\mathbb{C}^{(\boldsymbol{x})}$ is the closest cluster to $\boldsymbol{x}$ ($\mathbb{C}^{(\boldsymbol{x})} = \arg\min_{\mathbb{C}} \mathrm{dist}(\mathbb{C}, \boldsymbol{x})$) and $\mathrm{dom}(\mathbb{C})$ evaluates the dominant class of a cluster ($\mathrm{dom}(\mathbb{C}) = \arg\max_i \mathrm{m}(\mathbb{C})_i$, where $\mathrm{m}(\mathbb{C}) \in \mathbb{N}^L$ produces a cluster mapping vector describing the number of members of each class prediction). Table 1 summarizes the analysis results and confirms that both properties exist. Intuitively, a good feature extractor for classification purpose should produce similar features for all samples within the same class.

On the other hand, according to Tang et al. (2019), the existence of adversarial examples results from a mismatch between features used by human and those used by CNNs. Therefore, one intuitive approach for increasing CNN robustness is simply to drive CNN-learned features toward human-used features. However, it is impossible to understand and predict human-used features with any absolute degree of certainty. Thus, an alternative approach is to force the CNN-learned features to have some expected properties that human-used features should also have. For example, as mentioned above, features for the classification of objects belonging to the same class should be similar to one another.

Based on the above observations, the present study proposes a novel training process, designated as Manifold-Aware Training (MAT), for learning features which are both representative and compact. The experimental results confirm that models trained with MAT exhibit significantly higher robustness than existing state-of-the-art models. It would be clear later that our idea is, in some sense, similar to Pang et al. (2020) in that the authors proposed the Max-Mahalanobis Center (MMC) loss, which minimizes the distance of features to their assigned preset class centers. By showing that robustness of the model using MMC with adversarial training is higher than that using simply adversarial training, the authors claimed that high feature compactness results in locally sufficient

Table 1: Match rates of clean data and adversarial examples, which were generated using PGD and Decoupled Direction and Norm (DDN) attack (Rony et al., 2019), in naturally trained models (-N) and TRADES-trained models (-T).

| Data Type | MNIST-N | CIFAR10-N | MNIST-T | CIFAR10-T |
|-----------|---------|-----------|---------|-----------|
| Clean | 99.61% | 99.31% | 99.67% | 91.65% |
| PGD | 46.84% | 74.44% | 98.43% | 77.42% |
| DDN | 27.66% | 51.53% | 63.64% | 56.96% |

samples, which are beneficial for robust training according to Nakkiran (2019). Their conclusion of feature compactness helps robustness matches our idea.

The main contributions of our study can be summarized as follows:

- A better understanding of the relationship between robustness and the distance between adversarial examples and clean samples in the MAT-learned feature space.
- Our method improves the state-of-the-art performance from 57% to 80% for CIFAR10 and from 96% to 99% for MNIST in terms of robust accuracy.

The main notations used in present study are summarized in Appendix A

## 2 RELATED WORKS

One family of adversarial defense methods, referred to as adversarial-training-based methods, augments the training data with adversarial examples. For example, Zhang et al. (2019) defined the so-called TRADES loss, which is based on the trade-off between the clean accuracy (*i.e.*, the accuracy of clean images) and the robustness accuracy and has the form

$$\min_{\boldsymbol{\theta}} \frac{1}{N} \sum_{i=1}^{N} J(\boldsymbol{x}^{(i)}, y^{(i)}; \boldsymbol{\theta}) + \max_{\boldsymbol{x}'^{(i)} \in \mathcal{B}_\epsilon(\boldsymbol{x}^{(i)})} J'(\boldsymbol{x}^{(i)}, \boldsymbol{x}'^{(i)}; \boldsymbol{\theta})/\lambda, \tag{1}$$

where $\mathcal{B}_\epsilon(\boldsymbol{x}^{(i)})$ denotes the $l_p$-bounded ball ($\mathcal{B}_\epsilon(\boldsymbol{x}) = \{\boldsymbol{x}' \mid \|\boldsymbol{x}' - \boldsymbol{x}\|_p \leq \epsilon\}$), and $J(\cdot, \cdot; \boldsymbol{\theta})$ and $J'(\cdot, \cdot; \boldsymbol{\theta})$ are implemented using the cross-entropy loss and the Kullback-Leibler divergence criterion, respectively. Ding et al. (2020) proposed the Max Margin Adversarial (MMA) training method, which minimizes the cross-entropy loss for misclassified samples and maximizes the margin between them and the nearest decision boundaries for correctly classified samples. The authors proved that by minimizing the cross-entropy loss of the samples on the decision boundaries, the margin can be maximized. Wang et al. (2020) found that misclassified samples bring more robustness when used in adversarial training than correctly classified ones. Inspired by this observation, the authors proposed an objective function that encouraged stability around misclassified samples and optimized the classification loss for adversarial examples.

While the adversarial-training-based methods achieve state-of-the-art robust accuracy, they inherit some limitations (Nakkiran, 2019; Schmidt et al., 2018). Some studies try to alleviate these limitations to further improve the CNN robustness. For examples, according to the work of Schmidt et al. (2018), robust training requires significantly more samples than natural training. However, high sample density may make the training samples locally sufficient for robust training. Therefore, Pang et al. (2020) proposed the Max-Mahalanobis Center (MMC) loss as follows:

$$\mathcal{L}_{MMC}(\phi(\boldsymbol{x}), y) = \frac{1}{2} \|\phi(\boldsymbol{x}) - \boldsymbol{\mu}^{(y)}\|_2^2, \tag{2}$$

where $\phi(\cdot)$ is a CNN feature extractor, and $\boldsymbol{\mu}^y$ denotes the pre-set class center for class $y$. Additionally, the authors analyzed the limitations of traditional softmax and cross-entropy (SCE) loss based training methods regarding the induced sample density in the feature space. The authors then proved that the feature density induced by the MMC loss is guaranteed to be high around the centers ($\boldsymbol{\mu}^{(y)}$).

To our knowledge, however, improving the robustness of CNNs by manipulating the feature space attracts little attention. In view of this, we propose a novel defense method by training CNNs with

loss functions designed to induce feature compactness and improve the state-of-the-art robustness significantly.

## 3 PROPOSED METHOD

The proposed Manifold-Aware Training (MAT) algorithm is introduced in Subsection 3.1, and two additional loss functions to further improve the performance of MAT are described in Subsection 3.2. Moreover, a transformation technique designed to architecturally transform MAT-trained CNNs into traditional CNNs for compatibility with existing techniques is presented in Subsection 3.3.

### 3.1 MANIFOLD-AWARE TRAINING (MAT)

Recall the observed properties of features learned by traditional CNNs, *i.e.*, the non-clustering property and the confusing-distance property (see Section 1). This subsection introduces our proposed loss function to achieve intra-class feature compactness and inter-class feature dispersion. Specifically, the loss function minimizes the distance of CNN output features to their corresponding class centers while the class centers are far from one another. According to Pang et al. (2020), fixing class centers during the training phase makes better performance than updating them while learning for another objective (*e.g.*, intra-class feature compactness) since the CNN can focus on one objective rather than seeking a trade-off between multiple objectives. To ensure the pre-defined class centers are far from one another (to ensure inter-class dispersion), the distance between the samples is measured using the cosine-distance ($\mathrm{CD}(\boldsymbol{a}, \boldsymbol{b}) = 1 - \cos(\boldsymbol{a}, \boldsymbol{b})$, where $\cos(\boldsymbol{a}, \boldsymbol{b}) = (\boldsymbol{a} \cdot \boldsymbol{b})/(\|\boldsymbol{a}\|_2 \|\boldsymbol{b}\|_2)$) and the class centers are generated using the algorithm proposed by Pang et al. (2018), *i.e.*, the Max-Mahalanobis Distribution (MMD) centers. The generated center vectors are the vertices of an $L$-dimensional simplex such that the included angles between class center vectors are maximized, *i.e.*,

$$\{\boldsymbol{\mu}^{(1)}, ..., \boldsymbol{\mu}^{(L)}\} = \arg \min_{\boldsymbol{\mu}} \max_{i \neq j} \boldsymbol{\mu}^{(i)} \cdot \boldsymbol{\mu}^{(j)}, \|\boldsymbol{\mu}^{(k)}\|_2 = C \ \forall \ 1 \leq k \leq L. \tag{3}$$

Note that the details of the generation algorithm and examples of MMD centers are provided in Appendix B. Accordingly, the proposed loss function for intra-class compactness, *i.e.*, Feature To Center (FTC) loss, is defined as

$$\mathcal{L}_{FTC}(\boldsymbol{x}, y) = \mathrm{CD}(\phi(\boldsymbol{x}), \boldsymbol{\mu}^{(y)}). \tag{4}$$

In addition, in the inference phase, an input images $\boldsymbol{x}$ is predicted to belong to class $\hat{y}$ as:

$$\hat{y} = f(x) = \arg \min_{1 \leq k \leq L} \mathrm{CD}(\phi(\boldsymbol{x}), \boldsymbol{\mu}^{(k)}) = \arg \max_{1 \leq k \leq L} \cos(\phi(\boldsymbol{x}), \boldsymbol{\mu}^{(k)}). \tag{5}$$

### 3.2 AUXILIARY LOSS FUNCTIONS TO ENHANCE ROBUSTNESS

This subsection introduces two auxiliary loss functions which are proposed to further improve the robustness of MAT.

#### 3.2.1 SECOND-ORDER (SO) LOSS

Inspired by the approach proposed by Yan et al. (2018), this sub-section proposes an objective function which minimizes the magnitude of the gradient of the classification loss and thus improves the stability of the classification results. Generally speaking, backpropagation for a gradient value for an arbitrary CNN is difficult and expensive to compute. However, since the computation operations of the FTC loss after the feature layer are fixed, the gradient w.r.t the features can be computed simply by applying a fixed series of operations through forwarding. Note that rather than using an ordinary classification loss (*e.g.*, the cross-entropy loss), the FTC loss is used when applying this technique to MAT since the stability of the FTC loss implies both the stability of the output features and the classification performance. Therefore, the magnitude of the gradient of the FTC loss w.r.t the features can be formulated as the following Second-order (SO) loss, *i.e.*,

$$\mathcal{L}_{SO}(\boldsymbol{x}, y) = \|\nabla_{\boldsymbol{\phi}} \mathcal{L}_{FTC}\|_2^2 = \|\frac{1}{\|\boldsymbol{\phi}\|_2}[\frac{\boldsymbol{\mu}^{(y)}}{\|\boldsymbol{\mu}^{(y)}\|_2} - \cos(\boldsymbol{\phi}, \boldsymbol{\mu}^{(y)})\frac{\boldsymbol{\phi}}{\|\boldsymbol{\phi}\|_2}]\|_2^2. \tag{6}$$

The corresponding derivation is provided below as:

$$
\begin{aligned}
\nabla_{\boldsymbol{\phi}} \mathcal{L}_{FTC} &= \nabla_{\boldsymbol{\phi}}[1 - \cos(\boldsymbol{\phi}, \boldsymbol{\mu}^{(y)})] = -\nabla_{\boldsymbol{\phi}} \cos(\boldsymbol{\phi}, \boldsymbol{\mu}^{(y)}) \\
&= -\nabla_{\boldsymbol{\phi}} \frac{\boldsymbol{\phi} \cdot \boldsymbol{\mu}^{(y)}}{\|\boldsymbol{\phi}\|_2 \|\boldsymbol{\mu}^{(y)}\|_2} = -\frac{1}{\|\boldsymbol{\mu}^{(y)}\|_2} \nabla_{\boldsymbol{\phi}} \frac{\boldsymbol{\phi} \cdot \boldsymbol{\mu}^{(y)}}{\|\boldsymbol{\phi}\|_2} \\
&= -\frac{1}{\|\boldsymbol{\mu}^{(y)}\|_2}\Big[\frac{\|\boldsymbol{\phi}\|_2 \boldsymbol{\mu}^{(y)} - \boldsymbol{\phi} \cdot \boldsymbol{\mu}^{(y)} \frac{\boldsymbol{\phi}}{\|\boldsymbol{\phi}\|_2}}{\|\boldsymbol{\phi}\|_2^2}\Big] = -\Big[\frac{\boldsymbol{\mu}^{(y)}}{\|\boldsymbol{\phi}\|_2 \|\boldsymbol{\mu}^{(y)}\|_2} - \cos(\boldsymbol{\phi}, \boldsymbol{\mu}^{(y)}) \frac{\boldsymbol{\phi}}{\|\boldsymbol{\phi}\|_2^2}\Big] \\
&= -\frac{1}{\|\boldsymbol{\phi}\|_2}\Big[\frac{\boldsymbol{\mu}^{(y)}}{\|\boldsymbol{\mu}^{(y)}\|_2} - \cos(\boldsymbol{\phi}, \boldsymbol{\mu}^{(y)}) \frac{\boldsymbol{\phi}}{\|\boldsymbol{\phi}\|_2}\Big].
\end{aligned}
\tag{7}
$$

### 3.2.2 BOUNDED-INPUT-BOUNDED-OUTPUT (BIBO) LOSS

Conceptually, the adversarial example problem is a consequence of the local instability of CNNs. Specifically, the output of a CNN may change significantly when a small perturbation is added to the input. Intuitively, therefore, if the local instability is alleviated, the adversarial robustness should be correspondingly improved. This subsection thus proposes a Bounded-Input-Bounded-Output (BIBO) loss to minimize the difference in the feature space between samples with small perturbations and clean sample, *i.e.*,

$$
\mathcal{L}_{BIBO}(\boldsymbol{x}, y) = \max_{\boldsymbol{x}' \in \mathcal{B}_\epsilon(\boldsymbol{x})} \mathrm{CD}(\phi(\boldsymbol{x}), \phi(\boldsymbol{x}')).
\tag{8}
$$

Specifically, the maximization is practically achieved using PGD optimizationm, which was also adopted in the second term of the TRADES loss (Eq. 1).

### 3.2.3 TOTAL LOSS OF MAT

The total loss of MAT is defined by:

$$
\mathcal{L}_{MAT} = \mathcal{L}_{FTC} + \alpha \cdot \mathcal{L}_{SO} + \beta \cdot \mathcal{L}_{BIBO}.
\tag{9}
$$

The SO loss ensures the local stability of the feature distance to class center ($\mathcal{L}_{FTC}$) w.r.t the features while the BIBO loss ensures the local stability of the features w.r.t the input image. Therefore, the total loss stablizes the feature distance to class center when the input is subject to small perturbations.

### 3.3 MAT TRANSFORMATION

The MAT-trained models described above produce features rather than class scores (logits), as for traditional CNNs, and this difference may make the former incompatible with existing theories or techniques designed for traditional CNNs. For example, the objective functions of most adversarial attacks are based on logits, which MAT-trained CNNs do not produce. However, the MAT models can be transformed to be architecturally identical to traditional CNNs by using the cosine similarities to each class center as the logits. Additionally, since the $l_2$-norms of the center vectors are the same, the computation of the scores can be simplified to one of computing inner-products, as proved below:

*Proof.* Let $\boldsymbol{\mu}^{(i)}$ and $\boldsymbol{\mu}^{(j)}$ be center vectors of two different classes. We can derive:

$$
\cos(\boldsymbol{\phi}, \boldsymbol{\mu}^{(i)}) \lesseqgtr \cos(\boldsymbol{\phi}, \boldsymbol{\mu}^{(j)})
\tag{10}
$$

$$
\implies \frac{\boldsymbol{\phi} \cdot \boldsymbol{\mu}^{(i)}}{\|\boldsymbol{\phi}\|_2 \|\boldsymbol{\mu}^{(i)}\|_2} \lesseqgtr \frac{\boldsymbol{\phi} \cdot \boldsymbol{\mu}^{(j)}}{\|\boldsymbol{\phi}\|_2 \|\boldsymbol{\mu}^{(j)}\|_2} \implies \boldsymbol{\phi} \cdot \boldsymbol{\mu}^{(i)} \lesseqgtr \boldsymbol{\phi} \cdot \boldsymbol{\mu}^{(j)}.
\tag{11}
$$

$\square$

Note that the simplified logit computation can be achieved by adding a linear layer to the back end of the CNN with the weights set as the class centers and biases set as zero ($\boldsymbol{h} = \boldsymbol{W}\boldsymbol{\phi}, \boldsymbol{W} = [\boldsymbol{\mu}^{(1)}, \boldsymbol{\mu}^{(2)}, ..., \boldsymbol{\mu}^{(L)}]^T$, where $\boldsymbol{h}$ are the logits.)

Table 2: Robustness comparison. For Clean and PGD, the accuracy is reported. For C&W and DDN, the average perturbation norm of adversarial examples is reported.

| | | | CIFAR10 / MNIST | | | |
|---|---|---|---|---|---|---|
| Defense | SO | BIBO | Clean | PGD | C&W | DDN |
| Natural | - | - | 95.76% / 99.54% | 00.03% / 00.19% | 0.432 / 0.687 | 0.134 / 0.357 |
| AT | - | - | 78.49% / 99.30% | 45.93% / 88.26% | 0.770 / 0.755 | 1.054 / 2.545 |
| TRADES | - | O | 84.92% / 99.34% | 56.61% / 94.38% | 0.722 / 0.342 | 0.823 / 2.852 |
| **MAT** | X | X | 94.93% / 99.64% | 03.45% / 61.62% | 0.488 / 0.389 | 0.235 / 1.305 |
| | X | O | 85.04% / 99.34% | 53.58% / **99.30%** | 1.234 / **2.313** | 0.924 / **6.103** |
| | O | X | 95.21% / 99.68% | 79.92% / 82.06% | 0.576 / 0.418 | 0.595 / 1.570 |
| | O | O | 83.80% / 99.32% | **80.55%** / 99.28% | **2.457** / 2.063 | **2.750** / 4.723 |

# 4 EXPERIMENTAL RESULTS

This section evaluates the robustness of the proposed MAT algorithm. First, the assumptions made in the threat model are described in Subsection 4.1. In Subsection 4.2 and 4.3, the defense capability of our method is evaluated and compared with state-of-the-art, known as TRADES (Zhang et al., 2019) and MMC (Pang et al., 2020), respectively, along with verification against adaptive attacks in Subsection 4.4. Finally, we examine the effectiveness of feature compactness in Subsection 4.5.

## 4.1 THREAT MODEL

According to Carlini et al. (2019), defining threat models is essential to achieve fair and reliable evaluation and comparison. Since the motivation of our work is to test the worst-case robustnes, the goal of the adversary is assumed to be that of causing misclassification errors, *i.e.*, non-targeted adversaries. In pursuing this goal, the adversary is further assumed to have full knowledge of the victim model, including the architecture, parameters, and datasets used. In other words, white-box attacks were considered in this paper, where the perturbation of bounded attacks is constrained to have bounded $l_p$-norm while the mean $l_p$-norm of perturbation rather than robust accuracy is adopted for evaluation against unbounded attacks.

## 4.2 ROBUSTNESS OF MAT-TRAINED CNNS

This subsection compares the robustness of our proposed MAT with that of existing mechanisms. TRADES (Zhang et al., 2019) was selected as the state-of-the-art method because the authors released their code and trained models, which makes it easy to reproduce the experiments, and we note that both MMA (Ding et al., 2020) and MART (Wang et al., 2020) only yield comparable performance with TRADES. The basic adversarial training (AT) approach, which minimizes the cross-entropy loss for adversarial examples rather than clean samples, was chosen as a baseline. Furthermore, to perform an ablation study, all variants of MAT with various combination of loss functions were empirically compared to explore their capabilities in defending against attacks. The experiments use two datasets, namely MNIST and CIFAR10 (as in Zhang et al. (2019)), associated with VGG Net (Simonyan & Zisserman, 2015) and wide ResNet (Zagoruyko & Komodakis, 2016), respectively. A library called `foolbox` (Rauber et al., 2017) was utilized to construct the adversarial examples. Three common white-box attack methods were used for evaluation purpose: PGD (Madry et al., 2018), C&W (Carlini & Wagner, 2017), and DDN (Rony et al., 2019), where C&W and DDN belong to unbounded attacks. The parameters with respect to these attacks are shown in Appendix C. Before applying adversarial attacks to MAT-trained models, the models are transformed into traditional architecture using the approach described in Subsection 3.3. The robust accuracy under different attacks and the clean accuracy (the accuracy of clean test data) are shown in Table 2. We can observe that the MAT-trained CNN (with all loss functions applied) exhibit significantly higher robustness than TRADES[1].

---

[1]The second term in Eq. 1 is a BIBO-like loss, where the distance is measured by Kullback-Leibler divergence rather than cosine-distance.

Table 3: Classification accuracy (%) comparison with MMC (Pang et al., 2020) for CIFAR10. The superscripts **un** and **tar** denote untargeted and targeted PGD attacks, respectively, and the subscripts of PGD denote the number of iterations for conducting such attacks. Note that $\leq 1$ denotes the robust accuracy which is under 1%.

| Defense | Clean | $\epsilon = 8 / 255$ | | | | $\epsilon = 16 / 255$ | | | |
|---------|-------|----------------------|----------------------|----------------------|----------------------|----------------------|----------------------|----------------------|----------------------|
| | | $PGD_{10}^{tar}$ | $PGD_{10}^{un}$ | $PGD_{50}^{tar}$ | $PGD_{50}^{un}$ | $PGD_{10}^{tar}$ | $PGD_{10}^{un}$ | $PGD_{50}^{tar}$ | $PGD_{50}^{un}$ |
| Natural | 92.9 | $\leq 1$ | 3.7 | $\leq 1$ | 3.6 | $\leq 1$ | 2.9 | $\leq 1$ | 2.6 |
| AT | 80.9 | 69.8 | 55.4 | 69.4 | 53.9 | 53.3 | 34.1 | 38.5 | 21.5 |
| MMC | 81.8 | 70.8 | 56.3 | 70.1 | 55.0 | 54.7 | 37.4 | 39.9 | 27.7 |
| MAT | 83.8 | **80.1** | **80.4** | **76.1** | **80.4** | **79.0** | **79.9** | **60.4** | **79.8** |

Furthermore, as Carlini et al. (2019) stated, for bounded attacks, the accuracy-versus-perturbation-strength curve is more indicative of the robustness than the accuracy of the single perturbation strength. Therefore, the related curves are presented in Figure 2 (Note that the detailed values are presented in Appendix D).

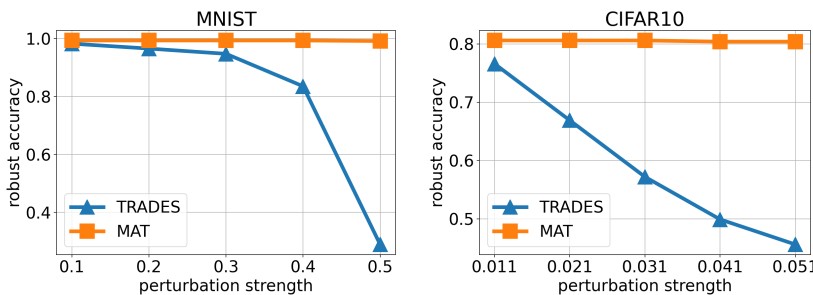

Figure 2: Robust accuracy vs. PGD with different perturbation strength ($\epsilon$). Note that here MAT employs the total loss function.

## 4.3 COMPARISON WITH MMC

As a similar approach to MAT, MMC (Pang et al., 2020) was chosen to be another baseline. To make fair comparison, the attack settings in MMC were used. The classification accuracies obtained from natural training (Natural), AT, and MMC were directly from Pang et al. (2020). Note that the model architecture for MAT was wide ResNet (Zagoruyko & Komodakis, 2016) while the one chosen in MMC was ResNet32. As shown in Table 3, the MAT model yields remarkably higher robustness than the MMC model, despite the underlying difference in model architectures.

## 4.4 ADAPTIVE ATTACKS

Carlini et al. (2019) stated that performing robustness evaluation against adaptive attacks, *i.e.*, attacks with objective functions which take the defense details into account, is essential to prevent a false sense of robustness. Thus, adaptive PGD attacks for MAT-trained models can be achieved by directly maximizing the training loss, including the regularization terms (SO loss or BIBO loss). Note that the BIBO loss is a result of maximization (using PGD). However, conducting attacks by optimizing an objective function incorporating such a maximized result suffers quadratic time complexity, which may be not realistic. Nonetheless, the inner maximization can actually be simplified as the inner cosine-distance function since the outer optimization is also a maximization process. The same technique can also be applied to the TRADES loss to conduct adaptive attack to TRADES models. In the experiments, the threat model of this adaptive attack is the same as described in Subsection 4.1. The robust accuracy under such attack of TRADES models is shown in Table 4. The best performance are still comparable with the state-of-the-art performance (of TRADES).

Table 4: Robust accuracy under the adaptive attack.

| Defense | SO | BIBO | CIFAR10 | | MNIST | |
|---|---|---|---|---|---|---|
| | | | PGD | Adaptive-PGD | PGD | Adaptive-PGD |
| TRADES | - | O | 56.61% | 56.26% | 94.38% | 93.06% |
| **MAT** | X | X | 03.45% | 00.45% | 76.79% | 09.78% |
| | X | O | 53.58% | 52.27% | 99.30% | 92.23% |
| | O | X | 79.92% | 01.27% | 82.06% | 09.79% |
| | O | O | 80.55% | 53.65% | 99.28% | 92.66% |

Table 5: Cosine-similarity to the predicted class center.

| Defense | SO | BIBO | CIFAR10 | | MNIST | |
|---|---|---|---|---|---|---|
| | | | Clean | PGD | Clean | PGD |
| **MAT** | X | X | 0.993±0.035 | 0.935±0.102 | 0.999±0.008 | 0.843±0.145 |
| | X | O | 0.879±0.123 | 0.615±0.068 | 0.995±0.023 | 0.727±0.020 |
| | O | X | 0.993±0.034 | 0.996±0.015 | 0.999±0.008 | 0.790±0.116 |
| | O | O | 0.845±0.133 | 0.569±0.075 | 0.996±0.021 | 0.762±0.055 |

Note that the adversarial training works (Zhang et al., 2019; Ding et al., 2020; Wang et al., 2020) did not consider adaptive attack evaluations, while an adaptive attack method for TRADES (Zhang et al., 2019) is proposed in our work, and it is intuitive that similar approach can be used for MMA (Ding et al., 2020) and MART (Wang et al., 2020) to construct adaptive adversarial examples. Additionally, Tramèr et al. (2020) have managed to conduct adaptive attacks toward MMC (Pang et al., 2020) models and reduce robust accuracy to under 0.5% by using the MMC loss as the objective function of PGD, which is the same as our strategy for MAT and TRADES.

### 4.5 Feature Clustering Performance by MAT

The effectiveness of MAT in performing feature clustering was examined by evaluating the cosine-similarity (mean $\pm$ standard deviation) to the class center of the predicted class. For each class, the mean and standard deviation were computed to quantify the level of clustering within class and over different classes, respectively. As shown in Table 5, the models with lower cosine-similarity for clean data have lower clean accuracy (Table 2) than the other models. However, the difference between their adversarial (PGD) similarity and the clean similarity is higher. Therefore, the cosine-similarity of clean data and the difference between the adversarial and clean similarity can be taken as indicators of the clean accuracy and robust accuracy, respectively.

## 5 Conclusion

To defend against adversarial attacks, this work analyzes the feature distribution of traditionally-trained CNNs for gaining more knowledge about adversarial examples. Two properties, *i.e.*, the non-clustering property and confusing-distance property, of the feature distribution are identified by means of t-SNE visualization and clustering analysis (showing the limitations regarding representativeness). In this paper, by exploiting feature compactness, a novel training process for improving model robustness, designated as MAT, is proposed, in which the FTC loss aims to induce intra-class feature compactness and inter-class feature dispersion, while two auxiliaries functions (the SO loss and the BIBO loss) of MAT are introduced. The experimental results show that the MAT-trained model with all loss functions applied exhibits significantly higher robustness than the state-of-the-art method-trained (TRADES) model. Additionally, the effectiveness of feature clustering and defense against adaptive attack are also evaluated. Our results reveal that the compactness of the clean features provides a useful indication of the clean accuracy, while the distance between the clean compactness and the adversarial compactness provides an indication of the robust accuracy.

Overall, this study gives a useful insight into the relation between feature compactness and robustness and provides a novel training process for improving the adversarial robustness of CNNs compared to current state-of-the-art methods.

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
