# OpenReview forum: "Manifold-aware Training: Increase Adversarial Robustness with Feature Clustering"
_ICLR.cc/2021/Conference — Reject_

### Official Review · AnonReviewer2 · 2020-10-29
**This paper proposes a manifold aware training strategy to learn compact features and improve the robustness of CNNs.**

**Rating:** 5
**Confidence:** 5

**Review:**

This paper proposes to leverage the manifold aware training to learn compact representation. The authors proposes to enforce the learned representation along with generated vectors for different clusters, which implicitly enlarge the margin of the prediction.
However the technical contribution as using the three-term loss to improve robustness is limited. In particular, it's is unclear what the equation (10) and (11) try to prove without a concrete theorem or lemma statement.

From the empirical performance, it looks promising from table 2 but it's also quite clear that the TRADES loss BIBO dominates the performance, and without adding this loss, the proposed MAT training cannot achieve high robustness. This is as expected and also render the proposed method less effective.
In addition, TRADES is evaluated on ImageNet and it would be good for the work to evaluate on ImageNet to demonstrate the generalization ability and scalability. It would also be good to explain why without the BIBO loss, the robustness against adaptive attack of MAT is almost 0 which again shows the weakness of the main proposed method.

It would also be necessary to provide analysis for the properties of the learned representation. For instance, if it is compact features, whether its rank is indeed lower, and whether the entropy of the learned features is indeed low in order to claim the consistent and compact feature representation.

---

### Official Review · AnonReviewer3 · 2020-10-30
**No theory, experiments only**

**Rating:** 4
**Confidence:** 3

**Review:**

Results: To defend against adversarial attacks, this work experimentally analyzes the feature distribution of traditionally- trained CNNs for gaining more knowledge about adversarial examples. Two properties, i.e., the non-clustering property and confusing-distance property, of the feature distribution are identified by means of t-SNE visualization and clustering analysis (showing the limitations regarding representativeness) in Figure 1. The authors introduce a loss function which separates out cluster centers of CNN output features, setting them as far as possible - so that model accuracy is preserved while strengthening robustness. They test on two datasets: CIFAR10, MNIST, and show improvements in "robustness" of the model.

Strong points: The experiments presented are promising in terms of increasing robustness of the learned models.

Weak points: Experiments are only conducted on two datasets, it's unclear how generalization these results are. Further, there is no theoretical development regarding manifolds in the feature space.

Minor typing errors:
"indication of the clean accuracy"
"using PGD optimizationm,"
"an input images x"

---

### Official Review · AnonReviewer1 · 2020-11-02
**Recommendation to Accept**

**Rating:** 7
**Confidence:** 3

**Review:**

Summary:
This paper tackles the problem of training models that are robust to adversarial inputs. The authors starts by observing that previous models generate embeddings that can both (i) place same-class embeddings in different clusters and (ii) different-class embeddings in close proximity. They then introduce new loss functions that penalize these behaviors and design a training procedure (MAT) around these new losses. Finally, they show favorable performance of MAT compared to state-of-the-art techniques for addressing adversarial robustness.

Reasons for score:
Overall, I vote for accepting. Training adversarially robust models is an important problem, and the paper’s experimental validation that the features of prior methods (TRADES) exhibit (i) non-clustering and (ii) confusing distance motivates the approach they take. The loss functions are explicitly designed to combat these issues, and the experimental results clearly show the favorability of the MAT procedure. In addition, the ablation study of the various components of the loss functions also adds some insight into the results. The paper is also very well written.

Cons:
It would be of interest to have some theoretical justification for the approach. Regarding the loss functions, it seems that BIBO should be a consequence of penalizing FTC loss and SO loss, and should not be explicitly needed (this is also somewhat consistent with the results of Table 2). Finally, some of the experimental results can be explored further. For example, in the ablation study, some of the experiments perform better without one of the loss functions, and it may help to explain such behavior.

Clarity / Typos:
The paper is very well written. A couple of minor points:
Feature compactness - Maybe explain this phrase better in the introduction (explained well in Section 3 introduction)
Eqn 1: Maybe write J(f(x), y) and J(f(x), f(x’)) instead

---

### Official Review · AnonReviewer5 · 2020-11-04
**The defense evaluation is not correct**

**Rating:** 1
**Confidence:** 5

**Review:**

This work proposes a defense that combines prior work on learning features that are compact for samples from the same but dispersed for samples from different classes (MMD by Pang et al.) with (a) a method to find better class centers, (b) a gradient-norm regularization and (c) an adversarial training regularization.

Unfortunately, the reported results on the robustness of the defense are clearly wrong. For one, the core part of this defense by Pang et al. was broken by [1] which is not mentioned here. More importantly, the adversarial attacks employed here are not suited for finding minimal adversarial perturbations against the proposed defense. This can be seen most clearly in Figure 2 (or Table 10 in the appendix): If we allow a perturbation with L-infinity norm of 0.5 on MNIST, then we can always find an adversarial perturbation simply by setting the whole image to a flat gray value of 0.5. In turn, any effective adversarial attack should drive network performance down to at least random baseline performance (10%) for epsilon = 0.5. Instead, the paper reports > 99% accuracy for this value under a PGD attack, which means that PGD is totally ineffective against the given defense and a very different adaptive attack would be needed to accurately measure its robustness. Similarly, in Table 3 the attack success of targeted attacks is often higher than for untargeted attacks, again a clear sign for ineffective attacks. The work also uses an adaptive attack which works better for some versions of MAT but performs similar to PGD in other cases. Hence, the adaptive attack employed here are not good enought.

The reason why the proposed attacks fail against the defense are probably simple: for one, the attacks optimise a different classificatioon loss then what is actually used by the model. Second, both auxiliary losses may give rise to gradient masking, the most common issue for gradient-based attacks to fail against a defense. I highly suggest the authors study [1] to get familiar with how to engineer strong adaptive attacks.

[1] On Adaptive Attacks to Adversarial Example Defenses, Florian Tramer, Nicholas Carlini, Wieland Brendel, Aleksander Madry, NeurIPS 2020, https://arxiv.org/abs/2002.08347

---

### Official Review · AnonReviewer4 · 2020-11-06
**AnonReviewer4 Review**

**Rating:** 5
**Confidence:** 3

**Review:**

# Summary

The authors propose a novel training process called Manifold-Aware
Training (MAT) to increase the robustness of the CNN against adversarial
examples. The authors compare MAT against the state-of-the-art in
defenses against adversarial evasion attacks (i.e., TRADES and MCC) and
show their approach outperforms it.

# Strengths

+  Interesting intuition of performing training "in the" manifold
+  Interesting intuition to support SO and BIBO losses

# Weaknesses

-  Lack of comparison with a similar approach
-  Lack of conclusive remarks / actionable points

# Comments

I praise the authors intuition of exploring the possibility of training
a classifier by exploiting knowledge of the manifold - its immediate
implication is that of focusing on lower dimensions of compact features
that would be more robust to manipulation (and thus adversarial
attacks). I also particularly appreciate the threat model and the fact
the approach is evaluated in a white-box setting, according to Carlini
et al. (2019). While the authors' intuition is interesting, I wonder how
easy this is to achieve in practice. In general, we have no knowledge of
the underlying manifold and I thus wonder what guarantees this approach
would provide. The results seem to show no clear loss-dependent trend
and I thus wonder whether we can easily draw conclusive remarks. (For
instance, should we use SO and BIBO always? From a theoretical
perspective, it seems so, but experiments seem to show otherwise.)

Figure 1 is interesting as it shows that the representative features of
same-class samples are not always similar to one another. Wasn't this
already explored in Szegedy et al.? Perhaps not visually, but the fact
that objects close in the input space get eventually separated in the
latent space across the layers of the CNN is quite known. Also, a
similar approach to the authors' proposal seems to be explored by Crecchi
et al. [1]. It would be interesting to compare and position MAT against
this.

## Additional Comments

In Section 3.2, the authors propose two auxiliary loss functions to
further improve the robustness of MAT. I wonder whether the BIBO loss
would just suffice for the purpose, instead of relying on the
second-order loss too. I appreciate the explanation in Section 3.2.3 but
it would be interesting to understand how one should expect to tune
alpha and beta accordingly.

Results on CIFAR10 seem less stable than compared to those on MNIST. In
particular, there is no trend that shows that relying on SO and BIBO on
a clean dataset provides better results than with a plain FTC loss:
94%->85%->95%->83%; why the 95%? Is that expected? Similar reasoning
can actually be applied to MNIST too when one looks at PGD:
61%->99%->82->99; why 82%? Is this expected? In contrast, TRADES seem to
show an expected trend (even when BIBO loss is considered).

The authors rely on the library 'foolbox' - my impression was that
cleverhans [2] represented the state-of-the-art when it comes to
experimenting with adversarial ML attacks. What advantages does foolbox
have compared to cleverhans?

Although off-topic for this work, it would be interesting to understand
whether MAT would be beneficial in defending against adversarial attacks
that consider realizable attacks (in the problem space). Figure 2 shows
the stability of MAT robustness for increasing values of perturbations.
Adversarial attacks in the problem-space might need to consider
additional constraints while being non-necessarily constrained in a
lp-norm [3].

[1] Crecchi et al. Detecting Adversarial Examples through Nonlinear
Dimensionality Reduction. ESANN 2019
(https://pralab.diee.unica.it/sites/default/files/crecchi19-esann.pdf)

[2] http://www.cleverhans.io/

[3] https://s2lab.kcl.ac.uk/projects/intriguing/ (IEEE S&P 2020)

### Minor Typos

"optimizationm" -> "optimization"

---

### Decision · Program_Chairs · 2021-01-07
**Final Decision**

**Decision:**

Reject

**Comment:**

Two reviewers expressed clear concerns about the paper but the authors did not provide any response.